# Copy Number Variation of the *SHE* Gene in Sheep and Its Association with Economic Traits

**DOI:** 10.3390/ani9080531

**Published:** 2019-08-06

**Authors:** Rui Jiang, Jie Cheng, Xiu-Kai Cao, Yi-Lei Ma, Buren Chaogetu, Yong-Zhen Huang, Xian-Yong Lan, Chu-Zhao Lei, Lin-Yong Hu, Hong Chen

**Affiliations:** 1Key Laboratory of Animal Genetics, Breeding and Reproduction of Shaanxi Province, College of Animal Science and Technology, Northwest A&F University, Yangling 712100, China; 2Animal Disease Control Center of Haixi Mongolian and Tibetan Autonomous Prefecture, Delingha 817000, China; 3Key Laboratory of Adaptation and Evolution of Plateau Biota, Northwest Institute of Plateau Biology, Chinese Academy of Sciences, Xining 810001, China

**Keywords:** sheep, *SHE* gene, copy number variation (CNV), growth traits, association

## Abstract

**Simple Summary:**

Src Homology 2 Domain Containing E (*SHE*) is a protein coding gene, and member of the SH2 domain-containing family. Sequencing revealed a 2000 bp copy number variation in the *SHE* gene. There is overlap between *SHE* copy number variation (CNV) and quantitative trait loci related to milk fat percentage and bone density. A total of 750 sheep, including Chaka sheep (CKS), Hu sheep (HS), Small Tail Han sheep (STHS) and Large Tail Han sheep (LTHS) were available to detect the CNV of the *SHE* gene and correlate these gene variations with economic traits. The results showed that there were more individuals with *SHE* copy number loss in CKS and HS than in STHS and LTHS. Association analysis showed that gain and normal copy number types performed better in body length (*p* < 0.05), circumference of cannon bone (*p* < 0.05), heart girth (*p* < 0.05), chest width (*p* < 0.05) and high at the cross (*p* < 0.05) in CKS, HS and STHS. Chi-square analyses found significant variation in the CNV of the *SHE* gene, so it varies greatly between varieties. These findings clarified the relationship between the CNV of the *SHE* gene and the economic traits in these four kinds of sheep, and provide a reference for sheep breeding.

**Abstract:**

Copy number variation (CNV) caused by gene rearrangement is an important part of genomic structural variation. We found that the copy number variation region of the Src Homology 2 Domain Containing E (*SHE*) gene correlates with a quantitative trait locus of sheep related to milk fat percentage and bone density. The aim of our study was to detect the copy number variation of the *SHE* gene in four sheep breeds and to conduct a correlation analysis with economic traits, hoping to provide some reference for sheep breeding. In this study, we examined 750 sheep from four Chinese breeds: Chaka sheep (CKS), Hu sheep (HS), Large Tail Han sheep (LTHS) and Small Tail Han sheep (STHS). We used qPCR to evaluate the copy number of the *SHE* gene, and then used general linear models to analyze the associations between CNV and economic traits. The results showed that there were more individuals with *SHE* copy number loss in CKS and HS than in STHS and LTHS individuals. Association analyses showed that gain and normal copy number types were correlated to body length, circumference of cannon bone, heart girth, chest width and high at the cross in CKS, HS and STHS (*p* < 0.05), but this association was not observed for LTHS. Chi-square values (χ^2^) found prominent differences in CNV distribution among the studied breeds. Overall, the CNV of the *SHE* gene may be an important consideration for sheep molecular breeding.

## 1. Introduction

A quantitative trait locus (QTL) is a region of DNA that is associated with a particular phenotypic trait, which could be attributed to the effects of multiple genes and their environment. QTL mapping has been widely used to identify the relationships between genetic markers and quantitative traits of interest. The mapping of QTL must use genetic markers, and one seeks to locate QTLs by looking for a link between the genetic marker and the quantitative trait of interest. In other words, the genetic marker and QTL are in linkage. Copy number variation could be an abnormality caused by genome rearrangement. Abnormally sized fragments can range from 50 bp to 1 Mb, and mainly arise by genome deletion, insertion, recombination and complex mutations at multiple chromosomal sites [1]. When the function of different genes in animals is missing or altered, this can lead to changes in the animal’s phenotype and disease susceptibility. Mutations can be indels, single nucleotide polymorphisms (SNPs) or copy number variation (CNV). Indels are insertions or deletions of bases in the genome of an organism, usually less than 50 bp in length. SNPs mainly refer to DNA sequence polymorphism caused by single nucleotide variation at the genome level and are more abundant in genomic variation. CNV has a larger variation area than indels and includes a larger total number of nucleotides than the number of SNPs, so CNVs may have greater influence on genomic sequences [2].

Sheep are raised all over the world, because their temperament is timid, docile and easy to domesticate, and they can provide products such as meat and fur for people. Four Chinese breeds of sheep were studied here, Chaka sheep (CKS), Hu sheep (HS), Large Tail Han sheep (LTHS) and Small Tail Han sheep (STHS). CKS is a crossbreed between crossing local crossbred sheep and Romney sheep. The HS, STHS and the LTHS are local breeds [3]. In recent years, there have been several studies on sheep CNV. In 2016, Zhu et al. [4] found that genes such as *PPARA*, *RXRA* and *KLF11* were involved in fat deposition and affected the tail type of sheep. Yang et al. [5] found that *BTG3*, *PTGS1* and *PSPH* genes overlapped with sheep CNV regions which were associated with fetal muscle development, prostaglandin (PG) synthesis and bone color, respectively.

The Src Homology 2 Domain Containing E (*SHE*) is a protein-coding gene which is located on sheep chromosome 1 and contains 6 exons, and encodes a protein with 495 amino acids whose molecular weight is 53950 Da. The *SHE* gene is an important member of the SH2 domain-containing family. The SH2 (Src homology 2) domain is a structurally conserved protein domain, found in the Src oncoprotein [6] and many other intracellular signal transduction proteins [7]. Proteins containing these domains can dock to phosphorylated tyrosine residues on other proteins, and these protein–protein interactions play essential roles in cellular growth and development. Therefore, the SH2 domain can have a great effect on the economic traits of animals [8]. Studies on the *SHE* gene have mainly focused on diseases, such as human diseases blepharoconjunctivitis and blepharitis [9]. There have been no reports analyzing the *SHE* gene in sheep. However, bioinformatics analysis shows that the *SHE* gene of sheep has 86.2% similarity with the human *SHE* gene, suggesting the sheep *SHE* gene may also participate in disease and cellular growth and development.

In this study, our aim was to verify the CNV of the *SHE* gene in the four sheep breeds of CKS, HS, STHS and LTHS, and then test the association between the CNV of the *SHE* gene and the economic traits of these four sheep breeds. In addition, we expected to find out whether copy number variation of the *SHE* gene had the same effect on the growth traits of these four sheep breeds. Finally, we hoped that this work can provide useful information for sheep molecular breeding.

## 2. Materials and Methods

### 2.1. Animal Welfare

All animal experiments in the president study were performed in conformity to the applicable guidelines, animal welfare laws and policies. Moreover, all experiments were approved by the Faculty Animal Policy and Welfare Committee of Northwest A&F University (FAPWC-NWAFU, protocol number, NWAFAC1008).

### 2.2. Animals and Growth Traits Measurements

We selected a total of 750 sheep from four kinds of sheep to study the copy number variation of the *SHE* gene: CKS (n = 302, Wulan country, Qinghai Province, China), HS (n = 198, Mengjin Country, Henan Province, China), LTHS (n = 61, Yongjing Country, Gansu Province, China) and STHS (n = 189, Yongjing Country, Gansu Province, China).

We measured the economic traits of these 750 sheep. For CKS, we measured body weight, height, length and heart girth. For HS, we measured body weight, height, slanting length, heart girth, rump length and circumference of cannon bone. In the LTHS population, body weight, height, slanting length, heart girth, circumference of cannon bone, high at the cross, chest depth and chest width were measured. In the STHS population, we measured body height, slanting length, heart girth, circumference of cannon bone, high at the cross, chest depth and chest width.

### 2.3. Preparation of Sample and Genomic DNA

We collected all the blood samples of the sheep and the genomic DNA was extracted using the method of phenol-chloroform as described in Sonstegard et al. [10]. The quantity of the genomic DNA samples was determined using a Nanodrop 2000 Spectrophotometer. DNA samples were sequenced at a concentration of 25 ng/μL per dilution and stored at −20 °C. None of the animals included in this study showed any adverse health conditions.

### 2.4. Candidate Gene Identification and Primer Design

We found that the *SHE* gene was located in QTLs related to milk fat percentage and bone density, and we identified a 2000 bp (NC_019458.2 103174001 bp–103176000 bp) copy number variation in the *SHE* gene (Figure 1) (Huang et al., unpublished data). This CNV overlaps with the QTL associated with sheep milk fat percentage and bone density (Figure 2). The *ANKRD1* gene appeared stable with two copies (Huang et al., unpublished data). We also found that the *ANKRD1* gene is present in two copies in both cattle and sheep according to the animal omics database, so we selected to use the *ANKRD1* gene as a diploid internal reference. All primers were designed by NCBI primer blast and are listed in Table 1.

### 2.5. Copy Number Variation and Gene Expression Analysis of the SHE Gene

We used quantitative polymerase chain reaction (qPCR) to determine the copy number of potential CNVs. Genomic DNA qPCR experiments were conducted using SYBR Green in triplicate reactions, each with a reaction volume of 10 μL, which contained 25 ng of genomic DNA, 5 μL of SYBR Premix Ex TaqII (GenStar, Beijing, China) and 0.5 μL of primers. Thermal cycling conditions were as follows: 95 °C for 10 min followed by 39 cycles of 95 °C for 15 s and 60 °C for 40 s. Melting curve analysis was completed at the end of the amplification with the following conditions: From 65 °C to 95 °C, 0.5 °C per 0.05 s. The copy number of the *SHE* gene was determined based on the assumption that the DNA fragment from the DNA of the calibrated animal has two copies.

### 2.6. Statistical Analysis

The copy number of the *SHE* gene was calculated using 2 × 2^−∆Ct^ [11]. Three types of copy number (gain, loss and normal) were classified as >2, <2 or 2 copies. We used a general linear model in SPSS v19.0 software (SPSS, Inc., Chicago, IL, USA) to examine the associations of *SHE* CNV with growth traits within CKS, HS, LTHS and STHS breeds, where age and sex were considered fixed factors. Individuals of each breed were unrelated and from the same farm and the phenotypes were measured in the same season [12]. We used the following model to analyze the CNV effects on phenotypic traits: Y_ijk_ = μ + A_i_ + S_j_ + CNV_k_ + e_ijk_, where Y_ijk_ is the observation of growth traits; μ is the overall mean of each trait; A_i_ is the effect due to age; S_j_ is effect due to sex; CNV_k_ is the effect of the *SHE* gene’s CNV type; and e_ijk_ is the random residual error. Finally, the variance between varieties was tested by chi-square [13]. The additive and dominance effect of the *SHE* gene CNV were calculated by the method of Falconer et al. [14] and Riollet et al. [15].

## 3. Results

### 3.1. The Distribution of SHE Gene’s CNV in Four Sheep Breeds

Available sequencing data revealed a 2000 bp copy number variation in the *SHE* gene. In order to study the distribution of this CNV in different sheep breeds, we examined CKS, HS, LTHS and STHS. The specific primers used are shown in Figure 3. Three types of copy number (gain, loss and normal) were classified as >2, <2 and 2 copies, and were calculated according to 2 × 2^−∆Ct^. As displayed in Figure 4A, the results revealed copy number variation of the *SHE* gene in the four breeds. The frequencies of copy number polymorphisms in the four sheep breeds revealed that most showed a loss copy number type, and this phenomenon was more pronounced in CKS and HS (Figure 4B).

### 3.2. Differences in CNV Distribution of the SHE Gene among Four Sheep Breeds

Based on the above findings, we calculated the frequencies of CNV types of the *SHE* gene in the four sheep breeds. The proportion of loss in CKS and HS breeds exceeded 50%, the proportion of loss in LTHS was only 10%, and there were relatively similar proportions of gain, loss and normal in STHS (Table 2). Chi-square values showed that a significantly different CNV distribution among the four sheep breeds exist (Table 3), indicating that the *SHE* CNV may be breed-specific.

### 3.3. Association between the Copy Number Variation and Growth Traits

Many recent studies found correlations of animal economic traits to copy number variation. Here, we analyzed the association of the *SHE* gene CNV copy number types with the economic traits of four sheep breeds using a general linear model. The association showed that the CNV of the *SHE* gene in CKS had a significant effect on body length (*p* < 0.05). In HS, CNV had a significant effect on the circumference of cannon bone and heart girth (*p* < 0.05). In STHS, the CNV of the *SHE* gene was related to chest width and high at the cross (*p* < 0.05). No effect was observed for LTHS (Table 4, Appendix A). Sheep with copy number gain had a significantly higher circumference of cannon bone and heart girth than those with loss or normal copy types in HS (*p* < 0.05). Consistently, gain and normal copy number subjects also had high body length, chest width and high at the cross than loss copy number groups in CKS and STHS, with a significance of *p* < 0.05. What is more, the phenotypic variance explained by this CNV is shown in Appendix A, and the additive, dominance genetic effects of this CNV are shown in Appendix A.

## 4. Discussion

CNVs are caused by the insertion, deletion or replication of large fragments. As the main genetic form of submicroscopic structural variation, CNVs are widely distributed in the human genome, and affect gene expression, phenotypic variation and adaptation by interfering with genes and changing gene dosage [16,17,18,19]. Recent studies have confirmed that CNVs are not only ubiquitous in humans, but are also widely interspersed in mammals and plants, including domestic cattle [20], dogs [21], chickens [22], pigs [23], goats [24], sheep [25] and rabbits [26]. The concept of QTL was proposed by Geldermann in 1975 [27] and refers to the localization of multiple genes that collectively control a trait to one or more fragments on a chromosome. For example, the CNV of the bovine *GBP4* gene is part of a QTL that determines its effect on the height of adult cattle [6].

As an SH2 domain-containing proteins, most studies of the *SHE* gene have focused on human diseases. The SH2 (Src Homology 2) domain is a structurally conserved protein domain present in the Src oncoprotein [7] and in many other intracellular signal-transducing proteins [28]. The function of this domain is to recognize the phosphorylated state of tyrosine residues, and initiates a series of events, which ultimately results in altered patterns of gene expression or other cellular responses. Therefore, the *SHE* gene in sheep is a promising candidate with an important function for sheep breeding.

We found that the CNV region of the *SHE* gene overlaps with a QTL region correlated with sheep milk fat percentage and bone density. Previous research reported that, in chickens, some QTL that correlated with CNV fragments are related to body weight, carcass weight and breast muscle mass [28,29]. Therefore, we speculated that the copy number variation of the *SHE* gene may be related to the economic traits of sheep, so four sheep (CKS, HS, STHS and LTHS) breeds were used in this study to analyze the CNV of the *SHE* gene. We found the CNV of the *SHE* gene existed in all four sheep breeds, and have associations with economic traits in CKS, HS and STHS. No association was observed in the LTHS, possibly due to the diversity across breeds [30]. For body length of CKS, the normal type was significantly better than the loss and gain types, and this difference was significant (*p* < 0.05). Notably, the gain type of HS associated with increased circumference of cannon bone and heart girth, and the normal and gain types were related with better chest width and high at the cross in STHS. Thus, we can conclude that the normal and gain CNV types of the *SHE* gene are related to better phenotypes in these four sheep breeds. We found different frequencies of the three CNV types in the four sheep breeds. Importantly, the dominant genotypes of these four sheep were not the same, which may be attributed to the diverse breeding backgrounds [31]. Gain and normal as the dominant CNV types of the *SHE* gene may be beneficial to the meat production and wool production traits of sheep, and further exploration is warranted.

CKS is a crossbred with little genomics studied. The meat value of CKS may be higher than the other local varieties. HS offers good quality lamb skin. LTHS have big tails, but poor development of the forequarters. STHS grow faster and are “high leg sheep” because of their tall and thin legs. The different characteristics of these four sheep breeds may be partially related to the CNV of the *SHE* gene.

In conclusion, this is the first detection and validation of CNVs of the *SHE* gene in four Chinese sheep breeds. The frequency of loss in CKS and HS exceeded 50%, the proportion of that in LTHS was only 10% and the proportion of the three copy number types in STHS was relatively balanced. Normal and gain copy number were associated with better economic traits than loss in CKS, HS and STHS, but had no difference in LTHS. The results suggested that the CNV of the *SHE* gene participates in economic traits and may be a molecular marker for sheep breeding.

## Figures and Tables

**Figure 1 animals-09-00531-f001:**
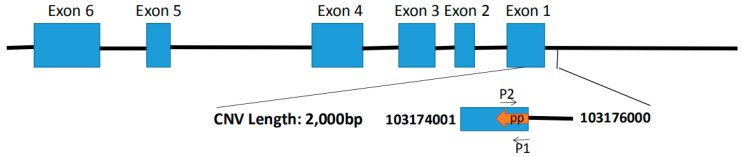
The region of the *SHE* gene’s copy number variation (CNV) in sheep breeds. The blue boxes indicate the coding area. The CNV region starts at 103174001 to 103176000 in Chr1 (Huang et al.). pp. (primer pair-CNV): 103174905 to 103175078, the detection sequence size is 174 bp.

**Figure 2 animals-09-00531-f002:**
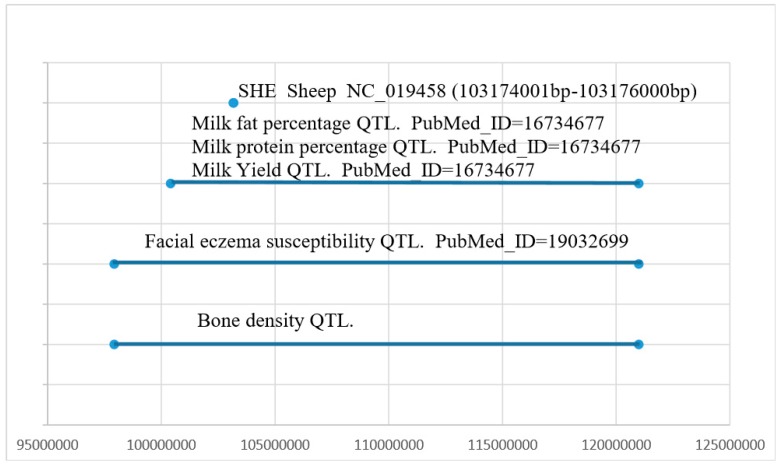
The CNV region of the *SHE* gene overlaps with the QTL of the sheep.

**Figure 3 animals-09-00531-f003:**
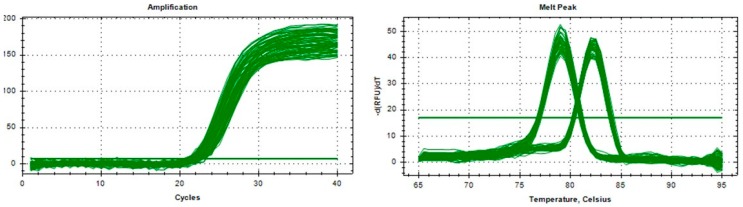
The specific primers of the *SHE* gene and the *ANKRD1* gene.

**Figure 4 animals-09-00531-f004:**
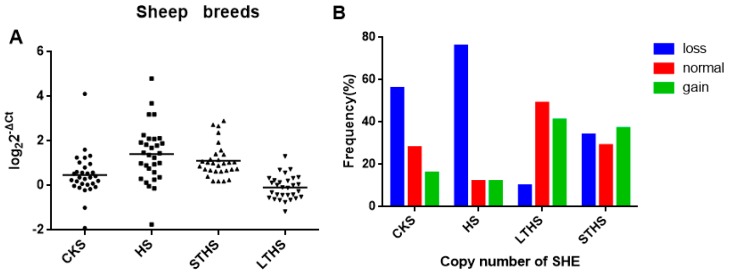
(**A**) The distribution of the *SHE* gene’s CNV in four sheep breeds. (**B**) The frequency of the copy numbers of the *SHE* gene’s CNV in the four sheep breeds.

**Table 1 animals-09-00531-t001:** The primer information used in this study.

Gene	Primer Pairs Sequences (5′–3′)	Amplification Length (bp)	Tm (°C)
*SHE*	1F: AACAAAGCGCATTTAGGGCA	174	60
	1R: ACGTCATGATCCAGCGATAGT	174	60
*ANKRD1*	2F: TGGGCACCACGAAATTCTCA	143	60
	2R: TGGCAGAAATGTGCGAACG	143	60

F: forward primer; R: reverse primer.

**Table 2 animals-09-00531-t002:** Frequency of different CNV types in four sheep breeds.

Breeds	Loss	Normal	Gain
CKS	0.56	0.28	0.16
HS	0.76	0.12	0.12
STHS	0.34	0.29	0.37
LTHS	0.10	0.49	0.41

Numbers represent the proportion of different CNV types in this breed.

**Table 3 animals-09-00531-t003:** The chi-square test of CNV types among Chinese sheep breeds.

Breed	CKS	HS	STHS	LTHS
CKS		22.886 (*p* = 1.07 × 10^−5^)	45.736 (*p* = 1.17 × 10^−10^)	33.844 (*p* = 4.47 × 10^−8^)
HS			84.897 (*p* = 3.67 × 10^−19^)	67.481 (*p* = 2.22 × 10^−15^)
STHS				15.809 (*p* = 3.69 × 10^−4^)
LTHS				

Chi-square values (χ^2^) for differences in CNV between two breeds.

**Table 4 animals-09-00531-t004:** Association analysis of the *SHE* gene CNV with growth traits in four sheep breeds.

Breed	Growth Traits	CNV Type (Mean ± SE)			*p*
		Loss	Normal	Gain	
CKS	Body length (cm)	71.66 ± 1.60 ^b^	74.22 ± 1.588 ^a^	73.33 ± 1.59 ^a,b^	0.028
HS	Circumference of cannon bone (cm)	76.39 ± 0.36 ^c^	76.40 ± 0.89 ^a,b^	78.77 ± 0.89 ^a^	0.046
STHS	Heart girth (cm)	7.03 ± 0.04 ^b^	7.29 ± 0.11 ^a,b^	7.31 ± 0.11 ^a^	0.012
	Chest width (cm)	19.58 ± 0.39 ^a,b^	20.26 ± 0.44 ^a^	18.60 ± 0.38 ^b^	0.019
	High at the cross (cm)	62.06 ± 0.51 ^B^	62.02 ± 0.58 ^B,C^	64.20 ± 0.50 ^A^	0.004

^a,b^ Values that differ significantly at *p* < 0.05; ^A,B,C^ Values that differ significantly at *p* < 0.01. CKS: loss (169), normal (86), gain (47). HS: loss (150), normal (24), gain (24). STHS: loss (65), normal (54), gain (70).

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
