# Peer review of "Copy Number Variation of the *SHE* Gene in Sheep and Its Association with Economic Traits"

_animals, 2019, doi:10.3390/ani9080531_

Round 1
Reviewer 1 Report
General comments:
In this study, the authors validated a copy number variation of the SHE Gene and performed association analyses with growth performance and body conformation traits in sheep. Although, the study is interesting, the following are my main comments:
- The authors stated in the manuscript that the CNV was identified in unpublished results (i.e. Huang et al., unpublished) but they did not provide a brief summary or enough information about the sequence analyses to detect the studied CNV. This information should be stated in briefly in the introduction and with more details in the material and methods sections.
- As the authors did not test the prediction ability of the identified CNV or any other genetic variants within SHE, how the authors conclude they are important for the breeding programs in of sheep?
- A total of 750 individuals with their records were available for testing the association between the CNV and the phenotypes, why the authors did not use the linear mixed model for the association analyses? did the authors have pedigree data for these animals? why the authors did not include the animal effect in the model?
- Did the authors examine the association within each breed or sheep population? if yes, why? Why the authors did not combine all the phenotypic records and fit breed as fixed effects?
- How much genetic and phynotypic variances explained by this CNV?
- Were the data collected from different farms? Did the authors include the farm effect in the model? if not why?
- The study did not provide information about additive or dominance genetic effects of this CNV? Thus, It would be nice to estimate the additive/dominance effect of the CNV.
Addressing the following comments and questions may improve the manuscript.
L3: the authors have measured some conformation traits, why the authors did not include in the title?
L20: - "milk fat percentage and bone density ", which species were these QTL detected? Probably in sheep, How are these QTL connected with the objectives of the current study? Please state the objective of the study clearly before stating material and methods as reported in the abstract section.
L22: "selected", were they selected or "were available" or "were collated' for the study? as the word "selected" indicated a selection bias?
L25: "normal copy number types performed better ", do the authors mean had significant associations? Please consider rephrasing. can the authors state estimate of its effects on these traits? what was the lowest p-value?
L34: “the cultivated variety”, Please consider rephrasing using "breed' or crossbred.
L37: "We used real-time quantitative PCR and general linear models to evaluate the copy number distributions ", please describe the methods clearly without combing the PCR with glm as they are two steps with different goals.
L38: "their effects on growth traits", do the authors mean "their associations"?
L42: "this correlation ", do the authors mean "this association"? did the authors measure correlation, LSM or substitution effects?
L43: "Overall, the CNV of the SHE gene may be an important consideration for the breeding of sheep", How did the authors come to this conclusion? Did the authors estimate how much variance explained in these traits by this CNV?
L51-52: “. By mapping one or more QTLs to genetic markers….”, Please consider rephrasing this sentence as the genetic markers are used to map or detect the QTL not the opposite. Did the authors mean “performing fine mapping” or “detecting new genetic variants within the confidence interval of the detected QTL”, please clarify in the text for the reader.
L59: Please review the definition of a SNP as I think the authors are confusing a "single nucleotide" with alternative nucleotide form or allele.
L60: SNPs are not structure variants, please consider rephrasing?
L63: Do the authors think these traits are due to selection, crossing and modern breeding programs occurring? or are these the general characteristics of sheep? Please delete the word "now".
L66: "variety", "cultivated" are plant breeding terms. Please rephrase and consider replacing "hybrid breeding variety" by "crossbred" and "cultivated by the mixing" by crossing X and Y sheep". Please apply across all the manuscript.
L67: Please replace “varieties” by “breeds” or “crossbred” according to their genetics.
L68: please add the citations of these several studies.
L68: “BTG3, PTGS1 and PSPH” please check the journal format for reporting gene names (i.e. full name or abbreviation, italic or not). Please apply across the entire manuscript.
L69: please replace “and” with "which" or "that", referring to these genes.
L70: please replace ";” by ".".
L70: The authors started with Yang et al 2018 then followed by Zhu et al 2016. Please cite based on year or rephrase to put more emphasis on the detected mechanisms if preferred.
L78: please add a citation.
L84: the objectives of the study should be stated clearly. For example, 1) validate the CNV, 2) Test the association between CNV and so on. Thus, method description should be moved to the material and method section.
L84: differences in what? Please state clearly.
L98: please consider splitting to avoid the “one sentence paragraph”.
L99: the full names of the breeds were stated previously so state only the abbreviations.
L103: replace "quantified" by "measured".
L103: - replace by growth performance.
- the authors have measured body conformation, are they one class within the growth traits in sheep?
L104: - delete the repeated word "body"
L110: please add the word "extraction and sequencing" after "genomic DNA".
L116: The authors have mentioned in the title only one gene, why the authors identify new one (i.e. ANKRD1 gene)? why did the authors start with a hypothesis about a particular CNV on chromosome 1?
L117: please check the journal policy for citing unpublished data? how do the reader track these results and information the authors are reporting here?
L120: "The ANKRD1 gene ", why the authors did not mention this gene prevoiusly?
L142: why the authors did not use the linear mixed model for the association analyses? do the authors have pedigree data for these animals? why the authors did not include the animal effect in the model? did the authors measure how much genetic and phynotypic variance explained by this CNV?
L145: "replace the word "analyze" by "test" or "examine"
L146: please replace "and" by "where".
L147: Did the authors include the farm effect in the model? why?
L147: "collected", do the authors mean "phenotypes were measured in the same season"? please consider rephrasing.
L182: "Correlation analysis ", Did the authors mean “association”?
Author Response
Response to Reviewer 1 Comments
Thank you very much for your valuable advice. I have explained it below, and modified and marked it in the manuscript.Point 1: The authors stated in the manuscript that the CNV was identified in unpublished results (i.e. Huang et al., unpublished) but they did not provide a brief summary or enough information about the sequence analyses to detect the studied CNV. This information should be stated in briefly in the introduction and with more details in the material and methods sections.
Response 1: Thank you for your advice. Since the study of Huang has not been published, we did not elaborate in detail in our manuscript, but the study showed that the QTL of SHE gene was located in a region related to growth traits, so we chose SHE gene for research. In order to make it easier for readers to understand, we explain it in the article. (L116-117)
Point 2: As the authors did not test the prediction ability of the identified CNV or any other genetic variants within SHE, how the authors conclude they are important for the breeding programs in of sheep?
Response 2: Firstly, SHE gene has been studied in human diseases Blepharoconjunctivitis and Blepharitis [1]. Secondly, we found that CNV region of SHE gene overlaps with QTL region of milk fat percentage and bone density, which can be used as candidate gene to study copy number variation. We know that genes are pleiotropic, for example Ran et.al found that some Femoral neck geometric parameters (FNGPs) are genetically correlated with Age at menarche (AAM). They performed a bivariate genome-wide association study (GWAS) to identify new candidate genes responsible for both FNGPs and AAM then found six SNPs that were associated with FNGPs and AAM. These SNPs are located in three genes (i.e. NRCAM, IDS and LOC148145), suggesting these three genes may co-regulate FNGPs and AAM [2]. So we speculate that SHE gene may be important to the breeding process of sheep based on the above studies. Your opinion is very good, and we will consider relevant verification in the subsequent experiments. [1] Cherif N, D'Hermies F, Barraco P, Elmaleh C, Renard G, Pouliquen Y., Meibomian adenocarcinoma in its blepharo-conjunctival form. Apropos of a case, J FR OPHTALMOL 20 (1997)293-6. [2] Shu Ran, Yu-Fang Pei, Yong-Jun Liu, Lei Zhang, Ying-Ying Han, Rong Hai, Qing Tian, Yong Lin, Tie-Lin Yang, Yan-Fang Guo, Hui Shen, Inderpal S Thethi, Xue-Zhen Zhu, Hong-Wen Deng., Bivariate genome-wide association analyses identified genes with pleiotropic effects for femoral neck bone geometry and age at menarche. PloS one 8 (2013) e60362.
Point 3: A total of 750 individuals with their records were available for testing the association between the CNV and the phenotypes, why the authors did not use the linear mixed model for the association analyses? did the authors have pedigree data for these animals? why the authors did not include the animal effect in the model?
Response 3: Your question is very meaningful. We can guarantee that there is no blood relationship within three generations and we explained it in the manuscript so we don't use a linear mixed model and didn’t include the animals effect in the model. (L146)
Point 4: Did the authors examine the association within each breed or sheep population? if yes, why? Why the authors did not combine all the phenotypic records and fit breed as fixed effects?
Response 4: Yes, we did. Considering the sheep breeds difference, that is the genetic background of different breeds are different, so we carried out the correlation analysis by variety.
Point 5: How much genetic and phynotypic variances explained by this CNV?
Response 5: We calculated genetic and phynotypic variances of this CNV and presented it in the following table. And as a supplementary figure Table S2 put in our manuscript. (L256-257) breeds Growth traits R2 percentage CKS body length (cm) 0.0004 0.04% HS circumference of cannon bone (cm) 0.0240 2.40% heart girth (cm) 0.0390 3.90% STHS chest width (cm) 0.0004 0.04% high at the cross (cm) 0.0388 3.9%
Point 6: Were the data collected from different farms? Did the authors include the farm effect in the model? if not why?
Response 6: Our data collected from four farms, but each variety comes from one farm, so we didn’t consider farm effect.
Point 7: The study did not provide information about additive or dominance genetic effects of this CNV? Thus, It would be nice to estimate the additive/dominance effect of the CNV.
Response 7: Thank you for your good suggestion. We calculated the additive and dominance effect of the CNV and as a supplementary figure Table S3 put in our manuscript. (L260-261) Supplementary Table S3. The additive and dominance effect of the SHE gene CNV. breeds Growth traits (cm) Type Additive effect value Dominant effect value CKS body length Loss -0.92 -0.31 Normal 0.61 0.72 Gain 2.14 -1.69 HS circumference of cannon bone Loss -0.16 0.08 Normal 0.28 -0.35 Gain 0.72 1.58 heart girth Loss -0.08 -0.01 Normal 0.14 0.04 Gain 0.36 -0.16 STHS chest width Loss 0.54 -0.62 Normal 0.02 0.58 Gain -0.51 -0.55 high at the cross Loss -1.14 0.60 Normal -0.55 -0.56 Gain 1.07 0.52
Point 8: L3: the authors have measured some conformation traits, why the authors did not include in the title?
Response 8: Thank you for your valuable advice. We chose an inappropriate term. We have replaced “Growth Traits” by “stature” in the article. (L3)
Point 9: L20: - "milk fat percentage and bone density ", which species were these QTL detected? Probably in sheep. How are these QTL connected with the objectives of the current study? Please state the objective of the study clearly before stating material and methods as reported in the abstract section.
Response 9: The Animal QTL Database shows the presence of the QTL of milk fat percentage on bovine chromosomes. No other species has detected the QTL of bone density. The existence of SHE gene QTL is one of the reasons why we chose this gene. Secondly, we speculated that bone density may also be related to weight, but we did not further study. According to your suggestion, we have stated the objective of our study in the abstract section. (L34-36)
Point 10: L22: "selected", were they selected or "were available" or "were collated' for the study? as the word "selected" indicated a selection bias?
Response 10: Thank you for your valuable advice. We have replaced “selected” by “available” in the article. (L23)
Point 11: L25: "normal copy number types performed better ", do the authors mean had significant associations? Please consider rephrasing. can the authors state estimate of its effects on these traits? what was the lowest p-value?
Response 11: Thank you for your advice. This sentence means had significant associations, we have indicated the P value in the article. (L26-27)
Point 12: L34: “the cultivated variety”, Please consider rephrasing using "breed' or crossbred.
Response 12: Thank you for your advice. We have replaced “the cultivated variety” by “crossbred” in all article. (L34, L68)
Point 13: L37: "We used real-time quantitative PCR and general linear models to evaluate the copy number distributions ", please describe the methods clearly without combing the PCR with glm as they are two steps with different goals.
Response 13: Thank you for your nice advice. We have revised the text. (L38-39)
Point 14: L38: "their effects on growth traits", do the authors mean "their associations"?
Response 14: Yes, I have corrected our test. (L39)
Point 15: L42: "this correlation ", do the authors mean "this association"? did the authors measure correlation, LSM or substitution effects?
Response 15: Yes, I mean associations, sorry for the misnomer, we didn’t measure correlation, LSM or substitution effects.
Point 16: L43: "Overall, the CNV of the SHE gene may be an important consideration for the breeding of sheep", How did the authors come to this conclusion? Did the authors estimate how much variance explained in these traits by this CNV?
Response 16: Thank you for your advice, according to the pleiotropy of the gene, we just assumed that the copy number variation of SHE gene was useful for sheep breeding. According to your Suggestions, we analyzed the influence of CNV of SHE gene on traits and presented it as Table S2 in our manuscript. (L255-256) Table S1. The genetic and phynotypic variances of SHE gene CNV breeds Growth traits R2 percentage CKS body length (cm) 0.0004 0.04% HS circumference of cannon bone (cm) 0.0240 2.40% heart girth (cm) 0.0390 3.90% STHS chest width (cm) 0.0004 0.04% high at the cross (cm) 0.0388 3.88%
Point 17: L51-52: “. By mapping one or more QTLs to genetic markers….”, Please consider rephrasing this sentence as the genetic markers are used to map or detect the QTL not the opposite. Did the authors mean “performing fine mapping” or “detecting new genetic variants within the confidence interval of the detected QTL”, please clarify in the text for the reader.
Response 17: Thank you very much for your advice. Since our mistake has brought inconvenience to the readers, we have corrected it in the manuscript. (L53-56)
Point 18: L59: Please review the definition of a SNP as I think the authors are confusing a "single nucleotide" with alternative nucleotide form or allele.
Response 18: Thank you for your advice. Correction has been made in the paper. (L60-62)
Point 19: L60: SNPs are not structure variants, please consider rephrasing?
Response 19: Thank you for your valuable advice. Correction has been made in the paper. (L62)
Point 20: L63: Do the authors think these traits are due to selection, crossing and modern breeding programs occurring? or are these the general characteristics of sheep? Please delete the word "now".
Response 20: Thank you for your advice. I think these are the general characteristics of sheep. Correction has been made in the paper. (L66)
Point 21: L66: "variety", "cultivated" are plant breeding terms. Please rephrase and consider replacing "hybrid breeding variety" by "crossbred" and "cultivated by the mixing" by crossing X and Y sheep". Please apply across all the manuscript.
Response 21: Thank you for your explanation. We have revised all the manuscript. (L34, L68)
Point 22: L67: Please replace “varieties” by “breeds” or “crossbred” according to their genetics.
Response 22: We have replaced “varieties” by “breeds” (L67)
Point 23: L68: please add the citations of these several studies.
Response 23: The citation had added in the manuscript. (L70)
Point 24: L68: “BTG3, PTGS1 and PSPH” please check the journal format for reporting gene names (i.e. full name or abbreviation, italic or not). Please apply across the entire manuscript.
Response 24: Thank you for your advice. These gene are abbreviation and should be italic, we have revised all the manuscript. (L71-74)
Point 25: L69: please replace “and” with "which" or "that", referring to these genes.
Response 25: I have replaced “and” with "which" in the manuscript. (L72)
Point 26: L70: please replace ";” by ".".
Response 26: Thanks for your reminding. I corrected it in the manuscript. (L72)
Point 27: L70: The authors started with Yang et al 2018 then followed by Zhu et al 2016. Please cite based on year or rephrase to put more emphasis on the detected mechanisms if preferred.
Response 27: Thanks for your advice. Correction has been made in our manuscript. (L70-73)
Point 28: L78: please add a citation.
Response 28: The citation had added in the manuscript. (L82)
Point 29: L84: the objectives of the study should be stated clearly. For example, 1) validate the CNV, 2) Test the association between CNV and so on. Thus, method description should be moved to the material and method section.
Response 29: Thanks for your good advice. According to your opinion, we have revised it in the manuscript. (L87-91)
Point 30: L84: differences in what? Please state clearly.
Response 30: The difference in the paper refers to whether the copy number variation of SHE gene has the same influence on the growth traits of these four sheep breeds. We have modified it in the manuscript. (L88-89)
Point 31: L98: please consider splitting to avoid the “one sentence paragraph”.
Response 31: Thanks for your advice. In this paragraph, we make a brief statement about animal welfare in two sentences. (L95-98)
Point 32: L99: the full names of the breeds were stated previously so state only the abbreviations.
Response 32: Your suggestion is very good. We have made some changes in the manuscript. (L100-103)
Point 33: L103: replace "quantified" by "measured".
Response 33: Correction has been made in the manuscript. (L103)
Point 34: L103: - replace by growth performance. - the authors have measured body conformation, are they one class within the growth traits in sheep?
Response 34: Your question is very good. The body conformation we measured are one class within the growth traits in sheep, information is available in the Animals QTL database. We've perfected the manuscript. (L104-109)
Point 35: L104: - delete the repeated word "body"
Response 35: Thanks for your advice. Correction has been made in the manuscript. (L103-108)
Point 36: L110: please add the word "extraction and sequencing" after "genomic DNA".
Response 36: Thanks for your advice. Correction has been made in the manuscript. (L111)
Point 37: L116: The authors have mentioned in the title only one gene, why the authors identify new one (i.e. ANKRD1 gene)? why did the authors start with a hypothesis about a particular CNV on chromosome 1?
Response 37: Thank you for your valuable advice. ANKRD1 gene appeared in the paper as an internal reference gene. This paper mainly studied copy number variation of SHE gene, and ANKRD1 gene was only used as an internal reference gene, so there was no relevant information introduced previously. The SHE gene is found on the sheep chromosome 1.
Point 38: L117: please check the journal policy for citing unpublished data? how do the reader track these results and information the authors are reporting here?
Response 38: Thank you for your advice. Since the study of Huang has not been published, we did not elaborate in detail in our manuscript, but the study showed that the QTL of SHE gene was located in a region related to growth traits, so we chose SHE gene for research. In order to make it easier for readers to understand, we explain it in the article. (L116-117)
Point 39: L120: "The ANKRD1 gene ", why the authors did not mention this gene prevoiusly?
Response 39: The same to Response 37. ANKRD1 gene was only used as an internal reference gene, so there was no relevant information introduced previously.
Point 40: L142: why the authors did not use the linear mixed model for the association analyses? do the authors have pedigree data for these animals? why the authors did not include the animal effect in the model? did the authors measure how much genetic and phynotypic variance explained by this CNV?
Response 40: We can guarantee that there is no blood relationship within three generations, so we don't use a linear mixed model and didn’t include the animals’ effect in the model. we analyzed the influence of CNV of SHE gene on traits and presented it as Table S2 in our manuscript. (L255-256) Table S1. The genetic and phynotypic variances of SHE gene CNV breeds Growth traits R2 percentage CKS body length (cm) 0.0004 0.04% HS circumference of cannon bone (cm) 0.0240 2.40% heart girth (cm) 0.0390 3.90% STHS chest width (cm) 0.0004 0.04% high at the cross (cm) 0.0388 3.88%
Point 41: L145: "replace the word "analyze" by "test" or "examine"
Response 41: Thanks for your advice. Correction has been made in the manuscript. (L145)
Point 42: L146: please replace "and" by "where".
Response 42: Thanks for your advice. Correction has been made in the manuscript. (L146)
Point 43: L147: Did the authors include the farm effect in the model? why?
Response 43: No, we didn’t include the farm effect, because each breed of sheep comes from the same farm.
Point 44: L147: "collected", do the authors mean "phenotypes were measured in the same season"? please consider rephrasing.
Response 44: Yes, the phenotypes were measured in the same season, correction has been made in the manuscript. (L147-148)
Point 45: L182: "Correlation analysis ", Did the authors mean “association”?
Response 45: Yes, we corrected it in the manuscript. (L182)
Reviewer 2 Report
The manuscript is interesting and scientifically sound. There are a few comments.
One major comment is the Huang’s study under submitting? If Huang's results are not published, it may be difficult to use in this study. The author may explain more details.
Line 57; “or disease” this may be missing. The author repeats “disease” twice.
Line 84; If the date has not been published, it may not need to put “(Huang et al., unpublished)”. Is this any means? I do recommend that the sequence date will be published in the NCBI database.
Line 84; qPCR should be “quantitative polymerase chain reaction PCR (qPCR)” at first appearance. Then remove the quantitative polymerase chain reaction in method section (Line 134).
Line 103 to 109; This is confusing for the reader. The author may rearrange common measurement and individual measurements. Also, the author indicates “the records of” in only STHS. Is this different from other species?
Line 111; Where did you collect the blood?
Line 111; Please describe DNA isolation as briefly. Sonstegard is not your co-author, so technically you do not know the exact technique.
Line 112; Nanodrop is photometer, so you can not assess the quality. It should be a quantity. Quality should be assessed by the electrophoresis methods, like BioAnalyser or agarose gel.
Line 117; if this is an unpublished date, the author needs to explain more details.
Line 130; Figure 2 is not clear enough.
Line 160; Figure 3 is not clear enough; we can not read the letter.
Author Response
Response to Reviewer 2 Comments
Thank you very much for your valuable advice. I have explained it below, and modified and marked it in the manuscript.
Point 1: One major comment is the Huang’s study under submitting? If Huang's results are not published, it may be difficult to use in this study. The author may explain more details.
Response 1: Thank you for your advice. Since the study of Huang has not been published, we did not elaborate in detail in our manuscript, but the study showed that the QTL of SHE gene was located in a region related to growth traits, so we chose SHE gene for research. In order to make it easier for readers to understand, we explain it in the article. (L116-117)
Point 2: Line 57; “or disease” this may be missing. The author repeats “disease” twice.
Response 2: Thank you for your advice. We have deleted “or disease” in the manuscript. (Line 59)
Point 3: Line 84; If the date has not been published, it may not need to put “(Huang et al., unpublished)”. Is this any means? I do recommend that the sequence date will be published in the NCBI database.
Response 3: I'm very sorry for the trouble. We put "(Huang et al., unpublished)" in the manuscript to show that our selection of gene is based on the basis, but I think your suggestion is very good, so we take your suggestion and remove "(Huang et al., unpublished)". Like point1, we have a simple explanation. (Line86-90)
Point 4: Line 84; qPCR should be “quantitative polymerase chain reaction PCR (qPCR)” at first appearance. Then remove the quantitative polymerase chain reaction in method section (Line 134).
Response 4: Thank you for your nice advice. According to the comments of reviewer 1, we have modified this part of the content. Here we have deleted “qPCR”. After carefully examining the full text, we determined that the first batch of “quantitative polymerase chain reaction PCR (qPCR)” appeared at Line34, followed by qPCR.
Point 5: Line 103 to 109; This is confusing for the reader. The author may rearrange common measurement and individual measurements. Also, the author indicates “the records of” in only STHS. Is this different from other species?
Response 5: Thank you for your advice. We made adjustments in the manuscript. There is no difference in four sheep breeds, we just want to avoid repeating the same phrase over and over again. And we have modified the manuscript to make it easier for readers to understand. (Line 104-109)
Point 6: Line 111; Where did you collect the blood?
Response 6: Your question is very necessary. We describe the animal's origin in line99-102.
CKS (n=302, Wulan country, Qinghai Province, China), HS (n=198, Mengjin Country, Henan Province, China), LTHS (n=61, Yongjing Country, Gansu Province, China), and STHS (n=189, Yongjing Country, Gansu Province, China).
Point 7: Line 111; Please describe DNA isolation as briefly. Sonstegard is not your co-author, so technically you do not know the exact technique.
Response 7: By reading Sonstegard's article, we extracted DNA using the method mentioned in the article. We will make a brief statement of the steps below:
We used the method of phenol-chloroform to separate DNA from whole blood, that is, proteinase K was used to dissolve proteins, Tris saturated phenol to denature proteins, chloroform to extract nucleic acids, and finally anhydrous ethanol to precipitate DNA.
What’s more we describe the DNA isolation as briefly in our manuscript. (Line112)
Point 8: Line 112; Nanodrop is photometer, so you can not assess the quality. It should be a quantity. Quality should be assessed by the electrophoresis methods, like BioAnalyser or agarose gel.
Response 8: Thank you for your advice. We corrected the mistake in the manuscript. (Line 113)
Point 9: Line 117; if this is an unpublished date, the author needs to explain more details.
Response 9: Your question is very necessary. We are very sorry for the inconvenience caused by our mistake. As shown by point1, we have explained accordingly.
Point 10: Line 130; Figure 2 is not clear enough.
Response 10: Thanks for your reminding. We have improved the clarity of Figure 2 in the manuscript. (Line130)
Point 11: Line 160; Figure 3 is not clear enough; we can not read the letter.
Response 11: We are very sorry for the inconvenience, we have improved the clarity of Figure 3 in the manuscript. (Line160)
Reviewer 3 Report
Thank you for this manuscript. It is well written and easy to follow and contributes valuable knowledge on the genetics of sheep.
Good use of figures.
This study is not in my field of expertise and yet it was relatively easy to follow.
I would like a bit more information in the conclusion on how this finding will affect sheep breeding. do we need to do genetic testing before breeding? Is this feasible? Or by selecting for high growth are we already inadvertently selecting for these genes already.
Author Response
Response to Reviewer 3 Comments
Point: I would like a bit more information in the conclusion on how this finding will affect sheep breeding. do we need to do genetic testing before breeding? Is this feasible? Or by selecting for high growth are we already inadve rtently selecting for these genes already.
Response: Your suggestion is very meaningful and we have been thinking about it.
1> Firstly, SHE gene has been studied in human diseases Blepharoconjunctivitis and Blepharitis [1]. Secondly, we found that CNV region of SHE gene overlaps with QTL region of milk fat percentage and bone density, which can be used as candidate gene to study copy number variation. We know that genes are pleiotropic, for example Ran et.al found that some Femoral neck geometric parameters (FNGPs) are genetically correlated with Age at menarche (AAM). They performed a bivariate genome-wide association study (GWAS) to identify new candidate genes responsible for both FNGPs and AAM then found six SNPs that were associated with FNGPs and AAM. These SNPs are located in three genes (i.e. NRCAM, IDS and LOC148145), suggesting these three genes may co-regulate FNGPs and AAM [2]. So we speculate that SHE gene may be important to the breeding process of sheep based on the above studies.
2> We think we can do genetic testing before breeding. We can carry out genetic testing before the birth of the sheep, and screen out the excellent individuals for continuous breeding, that is, marker-assisted selection. This will not only shorten the interval between generations but also saves costs, and provides some reference for the breeding of sheep.
[1] Cherif N, D'Hermies F, Barraco P, Elmaleh C, Renard G, Pouliquen Y., Meibomian adenocarcinoma in its blepharo-conjunctival form. Apropos of a case, J FR OPHTALMOL 20 (1997)293-6.
[2] Shu Ran, Yu-Fang Pei, Yong-Jun Liu, Lei Zhang, Ying-Ying Han, Rong Hai, Qing Tian, Yong Lin, Tie-Lin Yang, Yan-Fang Guo, Hui Shen, Inderpal S Thethi, Xue-Zhen Zhu, Hong-Wen Deng., Bivariate genome-wide association analyses identified genes with pleiotropic effects for femoral neck bone geometry and age at menarche. PloS one 8 (2013) e60362.
Round 2
Reviewer 1 Report
In the current version of the “Manuscript ID: animals-513846 - Revised Version”, the authors addressed most of the suggestions and comments. Nonetheless, there are few points that were not addressed as required. Thus, addressing these points (in bold) may improve the manuscript.
Point 4: Did the authors examine the association within each breed or sheep population? if yes, why? Why the authors did not combine all the phenotypic records and fit breed as fixed effects?
Response 4: Yes, we did. Considering the sheep breeds difference, that is the genetic background of different breeds are different, so we carried out the correlation analysis by variety.
=> So please replace “in CKS, HS, LTHS, and STHS breeds” by “within CKS, HS, LTHS, and STHS breeds” in Line 144. Please describe in the text that
- Point 5: How much genetic and phynotypic variances explained by this CNV?
Response 5: We calculated genetic and phynotypic variances of this CNV and presented it in the following table. And as a supplementary figure Table S2 put in our manuscript. (L256-257) breeds Growth traits R2 percentage CKS body length (cm) 0.0004 0.04% HS circumference of cannon bone (cm) 0.0240 2.40% heart girth (cm) 0.0390 3.90% STHS chest width (cm) 0.0004 0.04% high at the cross (cm) 0.0388 3.9%
- I think what you calculated is the difference in R2, is not it? This means the difference in phenotypic variance explained by the model with or without the CNV not the genetic variance. As you did not calculate the allele substitution effect, you cannot call it genetic variance? Especially these effects were not adjusted for animal effect.
- Please review and rephrase accordingly. Please review the Table S2 title as well.
Point 7: The study did not provide information about additive or dominance genetic effects of this CNV? Thus, It would be nice to estimate the additive/dominance effect of the CNV.
Response 7: Thank you for your good suggestion. We calculated the additive and dominance effect of the CNV and as a supplementary figure Table S3 put in our manuscript. (L260-261) Supplementary Table S3. The additive and dominance effect of the SHE gene CNV. breeds Growth traits (cm) Type Additive effect value Dominant effect value CKS body length Loss -0.92 -0.31 Normal 0.61 0.72 Gain 2.14 -1.69 HS circumference of cannon bone Loss -0.16 0.08 Normal 0.28 -0.35 Gain 0.72 1.58 heart girth Loss -0.08 -0.01 Normal 0.14 0.04 Gain 0.36 -0.16 STHS chest width Loss 0.54 -0.62 Normal 0.02 0.58 Gain -0.51 -0.55 high at the cross Loss -1.14 0.60 Normal -0.55 -0.56 Gain 1.07 0.52
- How did you calculate additive and dominance? Please explain in the text (i.e material and methods section).
Point 8: L3: the authors have measured some conformation traits, why the authors did not include in the title?
Response 8: Thank you for your valuable advice. We chose an inappropriate term. We have replaced “Growth Traits” by “stature” in the article. (L3)
- Can the word “stature” imply growth performance traits?
Author Response
Thank you very much for your valuable advice. Regarding your question, we explained the unclear points in the manuscript below (red) and reinterpreted them in our manuscript (blue).
Point 4: Did the authors examine the association within each breed or sheep population? if yes, why? Why the authors did not combine all the phenotypic records and fit breed as fixed effects?
Response 4: Yes, we did. Considering the sheep breeds difference, that is the genetic background of different breeds are different, so we carried out the correlation analysis by variety. => So please replace “in CKS, HS, LTHS, and STHS breeds” by “within CKS, HS, LTHS, and STHS breeds” in Line 144. Please describe in the text that
Round 2: Response4: Thank you for your nice advice. We have replaced “in CKS, HS, LTHS and STHS breeds” by “within CKS, HS, LTHS and STHS breeds” in the article. (L145)
Point 5: How much genetic and phynotypic variances explained by this CNV?
Response 5: We calculated genetic and phynotypic variances of this CNV and presented it in the following table. And as a supplementary figure Table S2 put in our manuscript. (L256-257) breeds Growth traits R2 percentage CKS body length (cm) 0.0004 0.04% HS circumference of cannon bone (cm) 0.0240 2.40% heart girth (cm) 0.0390 3.90% STHS chest width (cm) 0.0004 0.04% high at the cross (cm) 0.0388 3.9% -I think what you calculated is the difference in R2, is not it? This means the difference in phenotypic variance explained by the model with or without the CNV not the genetic variance. As you did not calculate the allele substitution effect, you cannot call it genetic variance? Especially these effects were not adjusted for animal effect. - Please review and rephrase accordingly. Please review the Table S2 title as well.
Round 2;Response 5:Yes, Thank you for your patient explanation and advice. We agree with your opinions very much and we have rephrased in the manuscript. (L190, Table S2: L257)
Point 7: The study did not provide information about additive or dominance genetic effects of this CNV? Thus, It would be nice to estimate the additive/dominance effect of the CNV.
Response 7: Thank you for your good suggestion. We calculated the additive and dominance effect of the CNV and as a supplementary figure Table S3 put in our manuscript. (L260-261) Supplementary Table S3. The additive and dominance effect of the SHE gene CNV. breeds Growth traits (cm) Type Additive effect value Dominant effect value CKS body length Loss -0.92 -0.31 Normal 0.61 0.72 Gain 2.14 -1.69 HS circumference of cannon bone Loss -0.16 0.08 Normal 0.28 -0.35 Gain 0.72 1.58 heart girth Loss -0.08 -0.01 Normal 0.14 0.04 Gain 0.36 -0.16 STHS chest width Loss 0.54 -0.62 Normal 0.02 0.58 Gain -0.51 -0.55 high at the cross Loss -1.14 0.60 Normal -0.55 -0.56 Gain 1.07 0.52 How did you calculate additive and dominance? Please explain in the text (i.e material and methods section).
Round 2: Response 7:Thank you for your advice. We used the calculation method of allele additive effect to calculate the additive effect value of SHE gene CNV, and the calculation method of allele dominant effect to calculate the dominant effect value of SHE gene CNV. And we have add description and quotes in the manuscript. (L151-152)
Point 8: L3: the authors have measured some conformation traits, why the authors did not include in the title?
Response 8: Thank you for your valuable advice. We chose an inappropriate term. We have replaced “Growth Traits” by “stature” in the article. (L3) - Can the word “stature” imply growth performance traits?
Round 2: Response 8:Thank you very much for your valuable advice. After thinking and literature we found "stature" is not particularly accurate, in order to include all the traits we measured, we finally use "economic traits" and we hope you can give us your comments. (L3)
Reviewer 2 Report
The revised manuscript has been correct according to the comments. This manuscript needs to be improved English quality.
Author Response
Dear reviewer:
Thank you very much for your valuable advice. Before submitting the manuscript, Dr. Kevin Li have modified the manuscript. According to your advice, we further modified and improved the language of our manuscript. The revised parts are marked in blue in the manuscript. We temporarily put the certificate of language editing at the end of the manuscript.